# Size-Dependent Liquid Crystal Behavior of Graphene Oxides for Preparation of Highly Ordered Graphene-Based Films

**Bongjin Chung** [1,2,†] , **Sunghwan Jin** [3,†] , **Junyoung Jeong** [1,2], **Giyoung Jeon** [1,2] **and Seongwoo Ryu** [1,2,*]

1 Department of Advanced Materials Engineering, The University of Suwon, 17, Wauan-gil, Bongdam-eup, Hwaseong-si, Gyeonggi-do 18323, Korea; bjchung8469@naver.com (B.C.); jepng3732@naver.com (J.J.); wjsrldud22@gmail.com (G.J.)
2 Center for Advanced Material Analysis, The University of Suwon, 17, Wauan-gil, Bongdam-eup, Hwaseong-si, Gyeonggi-do 18323, Korea
3 School of Energy and Chemical Engineering, Ulsan National Institute of Science and Technology (UNIST), Ulsan 44919, Korea; sunghwanjin00@gmail.com
* Correspondence: ryu@suwon.ac.kr; Tel.: +82-010-7367-0131
† These authors are equal major contributors.



**Featured Application: Graphene films with both excellent mechanical and electrical properties is considered as promising materials for multi-functional electronic devices: flexible sensors, wearable heaters, energy storage and conversion electrodes.**

**Abstract:** We report the preparation of a highly-oriented graphene-based film prepared from liquid crystal dispersion of graphene oxides (GOs). We observed that the liquid crystal behavior of GOs is highly affected by the lateral size of GO flakes: the critical concentration for liquid crystal formation decreased with the increase of the lateral size of GO flakes, which is in a good agreement with Onsager's theory. As a result, we were able to obtain highly-ordered graphene assemblies with large-sized GO flakes (150 ± 29 µm) at relatively low concentrations. By applying the shear force, we were able to obtain highly-oriented films from liquid crystal GO flakes. After hydrogen iodide (HI) reduction, GO films showed excellent mechanical strength and electrical conductivity, which were 278% and 283% higher, respectively, than those of films made from smaller sized GO flakes (28 ± 24 µm).

**Keywords:** graphene oxide; expanded graphite; liquid crystal; conductive film

## 1. Introduction

In recent decades, due to its excellent intrinsic mechanical properties (Young's modulus ≈ 1000 GPa and tensile strength ≈ 130 GPa) [1,2] and electrical properties (electrical conductivity of $10^7 \sim 10^8$ S/m) [3,4], graphene has received attention in lightweight structural and electrically conductive materials. Graphene is a two-dimensional nanomaterial with single atomic thickness, so it should be properly assembled at bulk scale for practical use. To realize a bulk graphene assembly, graphene oxides (GOs) have been widely used as starting materials. GOs have many functional groups and can be dispersed in various solvents [5,6]; because of this, a bulk graphene assembly can be produced by various solution processes such as vacuum filtration [7], wet spinning [8], casting [9], and the electrospray roll-to-roll process [10].

Currently, most graphene assemblies exhibit significantly lower properties than monolayer graphene due to the presence of defects, which are mainly located in the interconnection regions

between each graphene. To obtain a graphene assembly with better properties, defects within assemblies should be reduced by realizing more ordered graphenes. Recently, a nematic liquid crystal behavior of GOs in which GO flakes are aligned in a colloidal dispersion has been reported [6,11,12]. By subsequent solution processing of such dispersion, a highly-ordered graphene assembly can be obtained [13–15].

It is known that the liquid crystal behavior of GOs is highly affected by various factors such as concentration, lateral size and functional group of GO flakes [11,12]. More systematic studies are required to better understand the liquid crystal behavior of GOs and to obtain a more ordered GO assembly.

In this study, using liquid crystal GO dispersions, we developed bulk-scale graphene films (cm scale in lateral size with 3.96 μm thickness), which consisted of highly ordered graphene flakes. The liquid crystal dispersion of 150 ± 29 μm sized GO flakes showed a phase transformation of liquid crystal from isotropic to nematic at a concentration of 4 mg/mL, which is much lower than the liquid crystal dispersion of the 28 ± 24 μm sized GOs flakes (20 mg/mL). As a result, the larger-sized GO flakes showed a more ordered state than the smaller sized GO flakes at the same concentration. After film preparation from these liquid crystal dispersions and subsequent reduction by hydroiodic acid (HI), graphene-based films with larger-sized GO flakes showed better mechanical and electrical properties because there were fewer defects in the films.

## 2. Materials and Methods

### 2.1. Materials

Natural graphite (SP-1, Bay Carbon, Bay City, MI, USA), expandable graphite (Sigma Aldrich, St. Louis, MO, USA), sulfuric acid: $H_2SO_4$ (Samchun, Seoul, Korea), phosphoric acid ($H_3PO_4$, Junsei, Tokyo, Japan), potassium permanganate ($KMnO_4$, Sigma Aldrich, St. Louis, MO, USA), hydrogen peroxide ($H_2O_2$, Junsei, Tokyo, Japan), and hydrochloride (HCl, Samchun, Seoul, Korea) were used for the synthesis of graphene oxides (GOs). HI (Kanto Chemical, Tokyo, Japan) was used as a reducing agent of GOs. All chemicals were used as received.

### 2.2. Synthesis of Graphene Oxide (GO)

GO was synthesized by Hummer's method. One gram of natural graphite was added in 40 mL of $H_2SO_4$ at room temperature. The mixture was stirred magnetically at 300 rpm and cooled in an ice bath, followed by slow addition of 7 g of $KMnO_4$ for 5 min. Then, the mixture was kept at 35 °C on a hot plate for 2 h. The mixture was cooled in an ice bath and diluted by the addition of 200 mL distilled water. After dilution, $H_2O_2$ was added to the mixture to remove Mn ions until gas evolution was finished. The mixture was poured into a conical tube and centrifuged at 9000 rpm for 10 min. After removing the supernatant, 10% HCl aqueous solution was added to the precipitates to remove K ions. Then, the precipitates were re-dispersed in distilled water and centrifuged again at 12,000 rpm for cleaning. This process was repeated 3 times. After drying in a vacuum at 40 °C for 3 days, GO was obtained from the precipitates.

### 2.3. Synthesis of Large Sized Graphene Oxide (L-GO)

Large sized graphene oxide (L-GO) was also synthesized by Hummer's method, but an intercalation step using co-acids was added. One gram of expandable graphite was added to a mixture of 72 mL of $H_2SO_4$ and 8 mL of $H_3PO_4$ at room temperature. The mixture was stirred magnetically at 300 rpm and cooled in an ice bath. Then, 5 g of $KMnO_4$ was slowly added to the mixture within 5 min with magnetic stirring at 300 rpm for 3 h; the mixture was kept at 40 °C on a hot plate for 5 h. Other procedures were the same as those for GOs.

### *2.4. Preparation of Graphene-Based Films from Liquid Crystal GO Suspensions*

Graphene-based films were prepared from liquid crystal dispersions at concentrations of 4 to 20 mg/mL. These gel-like liquid crystal suspensions are stable even after 5 months of dispersion. Then, dispersions were poured onto the polytetrafluoroethylene (PTFE) substrates, and were reoriented by a doctor blade, with a blade speed of 50 mm/s, to obtain films having horizontally aligned flakes. Finally, GO and L-GO liquid crystal (LC) spreads were dried in a vacuum at 40 °C for 2 days.

### *2.5. Chemical Reduction of Graphene-Based Films*

The dried GO and L-GO films were reduced by HI [16–18]. GO or L-GO films were dipped in 10 mL of HI (55% in water) for 15 min; GO films floated on the surface of HI as a result of reduction. The films were turned upside down and floated on the surface of HI for an additional 15 min. After that, the films were taken out and washed with distilled water. This cleaning process was repeated 3 times. Then, the films were dried in a vacuum at 40 °C for 1 day.

### *2.6. Characterization*

Material characterization was supported by the Center for Advanced Materials Analysis (The University of Suwon). Scanning electron microscopy (SEM; APREO, FEI, Hillsboro, OR, USA) was used to observe the size of the GO and L-GO flakes, and the thickness of the GO and L-GO films. Chemical compositions of GO and L-GOs film were determined by Fourier-transform infrared spectroscopy (FTIR; FT-IR spectrum Two, Parkin Elmer, Waltham, MA, USA), X-ray photoelectron spectroscopy (XPS; K-alpha Plus, Thermo Fisher Scientific, Waltham, MA, USA), confocal Raman microscopy (FEX, Nost, Gyeonggi, Korea), and UV-Vis-NIR spectrophotometry (Lambda 750, Perkin Elmer, Waltham, MA, USA). X-ray diffraction (XRD; ARL EQUINOX 3000, Thermo Fisher Scientific, Waltham, MA, USA) was used to detect d-spacing. Polarizing microscopy (ECLIPSE LV100ND POL, Nikon, Tokyo, Japan) was used to observe the liquid crystal behaviors of GO and L-GO, and their shear viscosity was measured by a rheometer (MCR 300, Paar Physica, Graz, Austria). Before the measurement, pre-shear of 500/s was applied for 60 s to ensure homogeneous dispersion of suspension. Between shear rates ranging from 20 to 760/s, the viscosity of each GO suspension was measured in a stepwise manner for 40 min. Tensile strength was investigated using a microforce testing machine (MicroTester 8848, Instron, Norwood, MA, USA), and tensile tests were on rectangular samples with a 16 mm gauge length at a rate of 2.5 mm/min. The sheet resistance of films was measured with a 4-point probe system (CMT-SR1000N, Advanced instrument technology, Cumming, GA, USA). The cross-sectional area of the film was precisely evaluated from the cross-sectional SEM image using an image analysis program (Image J, NIH, Bethesda, MD, USA, 2020). The electrical conductivity of the films was obtained by dividing the sheet resistance into the cross-sectional area.

## 3. Results and Discussion

We used Hummer's method to produce GOs but introduced an intercalation step using co-acids to obtain larger-sized GO (L-GO) flakes. Graphite was immersed in the co-acid solution of $H_2SO_4$ and $H_3PO_4$, which resulted in an interaction of $H_2SO_4$ and $HSO_4^-$ with graphenes in HOPG. This acid-based intercalation−oxidation chemistry of graphite is well known: $H_3PO_4$ can be intercalated effectively between graphenes, and $H_2SO_4$ predominantly causes the oxidation of graphenes [19,20]. In addition, co-acid molecules form hydrogen bonds between them, which increases the intercalation distance between graphite layers. As a result, it was reported that the interlayer spacing of graphite (3.35 Å) increases significantly over ~8 Å [19], thereby decreasing the interlayer interactions between graphenes, as schematically shown in Figure 1a. After preparation of GOs with (and without) the intercalation step using co-acids, and their subsequent exfoliation in waters, the lateral size of L-GO flakes was 150 ± 29 μm (measured by 51 flakes), whereas that of GO flakes was 28 ± 24 μm (measured

by 65 flakes; Figure 1b), although both exfoliations were performed using the same procedure (see experimental section).

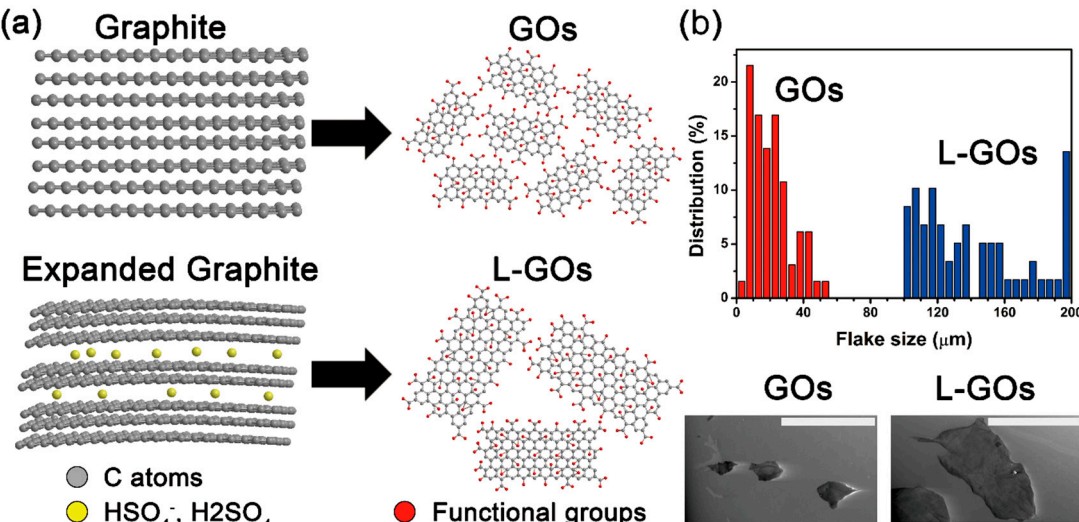

**Figure 1.** (**a**) Schematics for preparation of graphene oxides (GOs) and larger-sized GOs (L-GOs) from graphite (top) and expanded graphite (bottom), respectively; (**b**) distribution of lateral flake sizes of GOs and L-GOs (top) and their scanning electron microscopy (SEM) images (bottom). Scale bars in SEM images are 100 μm.

Liquid crystal behaviors of GO suspensions with different sizes have been systematically studied by changing their concentrations. In general, as the GO concentration increases, overlap of excluded volumes of GO flakes occurs; therefore, random movement of GO flakes in the suspension is limited. As a result, the arrangement of GO flakes in the suspension changes from random isotropic phase to ordered nematic phase [21]. This liquid crystal phase transition is known to be strongly affected by the size of colloidal particles. According to Onsager's theory [22], the empirical value of the critical concentration has the relationship per the following equation:

$$\Phi \approx 4T/W, \tag{1}$$

where $\Phi$ is the volume fraction of colloidal particles in the suspension, and T and W are the thickness and lateral width of the circular particles, respectively. GOs are also colloidal disks in a suspension; so, as the lateral size of the GO flakes increases, the critical concentration for the formation of liquid crystals decreases. To track the liquid crystal behaviors of GO and L-GO suspensions, we employed polarized optical microscopy (POM). Observation of the birefringent domains in the POM image can reveal the ordering of the GO flakes in the suspension [12]. Figure 2a,b show POM images of the GO and L-GO suspensions with increasing concentrations. For the GO suspension, as the concentration increased, we observed the appearance of isotropic circular domains at the concentration of 12 mg/mL and ordered lamellar domains at 20 mg/mL. This is a common characteristic of lyotropic liquid crystal materials: as the concentration of colloidal particles increases, a more ordered liquid crystal phase is formed [23,24]. In contrast, for the L-GO suspension, ordered lamellar domains began to appear at the concentration of 4 mg/mL. Compared to the GO suspension, the critical concentration for the ordered lamellar liquid was four times lower for the L-GO suspension. To further investigate the liquid crystal behavior of GO and L-GO suspension, we examined the shear rate dependencies of the viscosity for these suspensions at different concentrations. The GO suspensions showed almost constant viscosity over a wide range of shear rates when their concentrations were 4 and 12 mg/mL. However, when the concentration increased to 20 mg/mL, the viscosity decreased as the shear rate increased. Such viscosity change is known as shear thinning, which has been generally observed in nematic liquid

crystal suspension [25]. For the L-GO suspension, clear shear-thinning behavior was observed even at the lowest concentration of 4 mg/mL, which suggested that L-GO suspensions could form the nematic liquid crystal phase at a much lower concentration than the GO suspension.

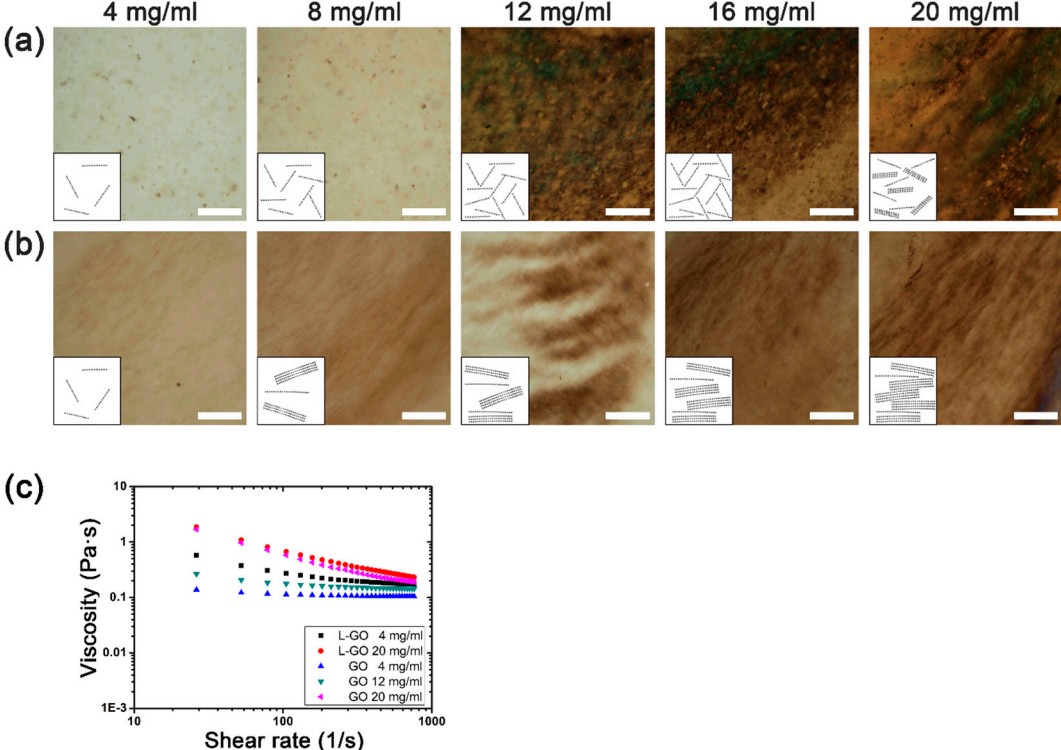

**Figure 2.** Polarized optical microscopy (POM) images of (**a**) GO and (**b**) L-GO suspensions according to their concentrations. Insets show schematics of GO and L-GO flakes in suspensions for each POM image. Scale bars are 200 μm. (**c**) Plots of viscosity versus shear rate according to concentrations of GO and L-GO suspension.

Our observations indicated that this difference could mainly be attributed to the average flake size of the L-GOs, which is about four times larger than that of the GOs. In good agreement with Onsager's theory, the four times larger GO flakes induced a four times lower critical concentration for formation of the liquid crystal. It was reported that the liquid crystal behavior of GO suspensions is also influenced by the surface charge of the GO flakes [12]. However, the zeta potentials of the GO (−31.64 ± 2.54 mV) and L-GO (−32.96 ± 2.07 mV) suspensions showed similar values, suggesting that the surface charges do not contribute to the difference in critical concentrations. In addition, lamellar domains in L-GO increased as the concentration of L-GO increased, and more continuous lamellar domains formed, like a hairy structure. As a result, the L-GO suspension showed a more ordered state than that of the GO suspension at the same concentration.

Using the simple doctor blade technique, bulk-scale GO films were prepared from the GO and L-GO suspensions at a concentration of 20 mg/mL (Figure 3a). At this concentration, both GO and L-GO suspensions showed ordered nematic liquid crystal behavior; therefore, aligned GO flakes in the films were easily obtained by applying shear forces with the doctor blade (Figure 3b). The thicknesses of both the GO and L-GO films were around 30 μm and many micrometer scale voids were observed in their cross-sections. After HI reduction, these microscale voids disappeared, and film thicknesses were reduced to around 3.96 μm. These observations were consistent with our XRD results (Figure 3c). Due possibly to the removal of oxygen functional groups or remaining intercalants between GO flakes, the d-spacings of GO (9.40 Å) and L-GO (9.88 Å) films were remarkably reduced to 3.72 and 3.68 Å, respectively, by HI reduction.

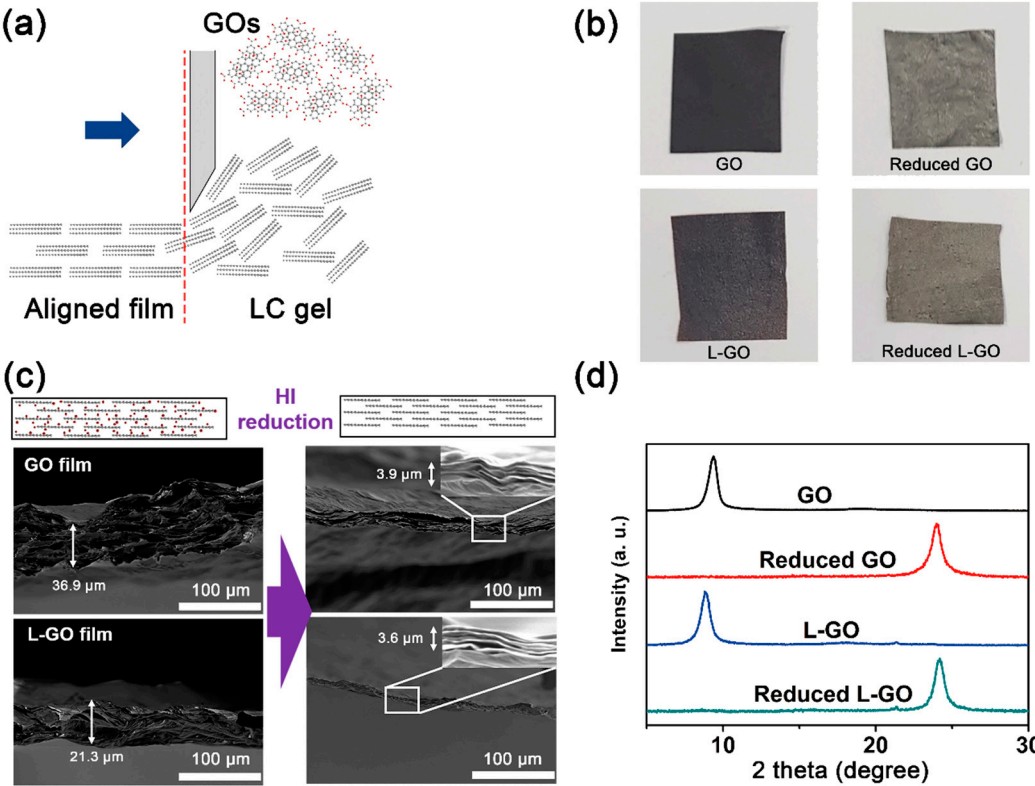

**Figure 3.** (**a**) Schematic of preparation of graphene-based films from liquid crystal GO suspensions. Gray and red circles indicate carbon atoms and functional groups, respectively. (**b**) Photograph of GO and L-GO films before and after reduction. (**c**) Structural change in graphene-based films by HI reduction: schematic illustration (top), and cross-sectional SEM images (bottom). Scale bars are 100 μm, and insets are high magnification (×10,000) images of the regions marked with white rectangles. (**d**) XRD spectra of GO and L-GO films before and after hydrogen iodide (HI) reduction.

The chemical structures of graphene-based films made from GO and L-GO flakes were analyzed by FTIR spectroscopy, X-ray photoelectron spectroscopy (XPS), and ultraviolet–visible (UV-Vis) spectroscopy. The FTIR spectra of both GO and L-GO showed five typical peaks at 1050, 1130, 1620, 1720, and 3320 $cm^{-1}$, which corresponded to the C–O, C–O–C, C=C, and C=O stretching and O–H vibration modes, respectively (Figure 4a). Compared to GO, L-GO showed weaker O–H vibration and C=O stretching peaks; it also showed a slightly stronger C–O–C stretching peak. These differences may have resulted from size differences between the GO and L-GO flakes. It is known that oxygen functional groups on the basal planes of GOs are mainly epoxides, whereas carboxylic functional groups are most abundant at the edges of GOs [26–28]. Increasing the flake size means that the fraction of edges in the overall flakes decreases, resulting in a decreased fraction of carboxylic functional groups and an increase of epoxide groups. This observation is in good agreement with the XPS results (Figure 4b): the C1s spectrum of the GOs deconvoluted to 37.7 at % of C–C, 52.8 at % of C–O, and 9.5 at % of C=O bonds, whereas L-GO showed 34.6 at % of C–C, 57.4 at % of C–O, and 8.0 at % of C–O bonds. The ratio of C=O/C–C was 1.4 and that of C–O/C–C was 0.3 for GOs; these values were 1.7 and 0.2 for the L-GOs, respectively, which indicated that L-GO flakes have more epoxide functional groups and fewer carboxylic functional groups than GO flakes. After HI reduction, due to the removal of oxygen functional groups and restoration of graphitic structures during chemical reduction, the intensities of the C–O and C=O bonds dramatically decreased and that of the C–C bonds increased for both the GO and L-GO films [29].

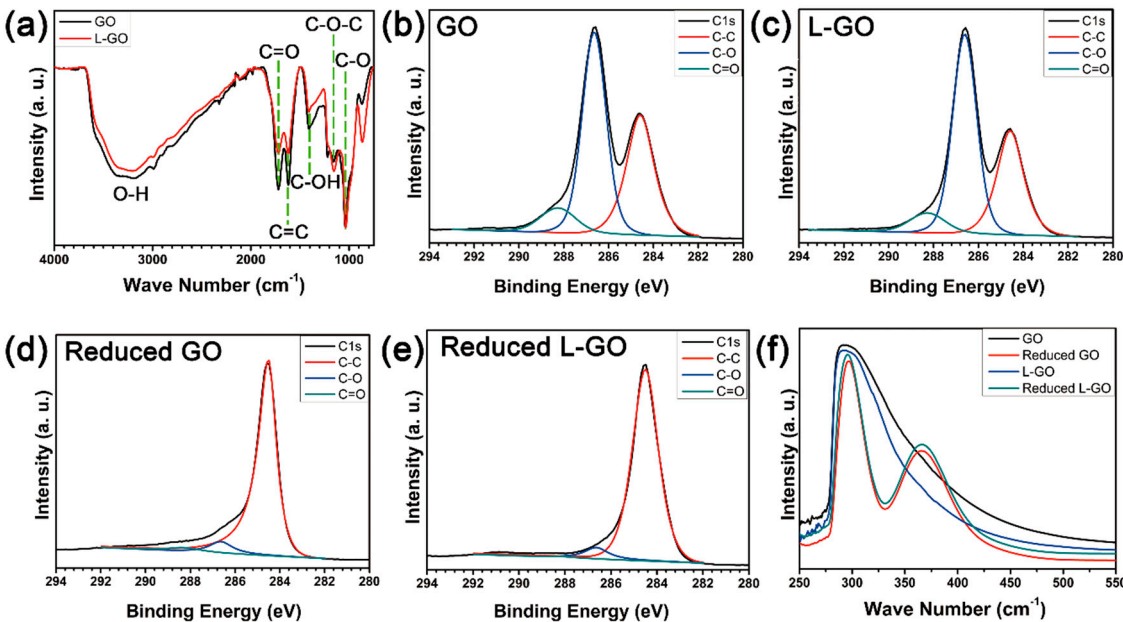

**Figure 4.** (**a**) FTIR spectra of GO and L-GO films. XPS C1s spectra: (**b**) GO, (**c**) L-GO, (**d**) reduced GO, and (**e**) reduced L-GO. (**f**) UV-VIS absorption spectra of GO and L-GO films before and after reduction.

The mechanical and electrical properties of the GO and L-GO films were evaluated (Figure 5a,b). The L-GO film exhibited 1.32× higher modulus (3.7 GPa), 1.98× higher ultimate tensile strength (20.37 MPa), and 1.28× higher elongation at break (0.55%) than the GO film (modulus: 2.8 GPa, ultimate tensile strength: 10.28 MPa, elongation at break: 0.43%). These mechanical properties of both the GO and L-GO films were greatly improved by HI reduction: 8.3 GPa modulus, 130.76 MPa ultimate tensile strength, and 3.09% elongation at break for the L-GO film, and 5.7 GPa modulus, 47.00 MPa ultimate tensile strength, and 1.53% of elongation at break for the GO film were obtained. The reduced L-GO film also exhibited only 37% (8.2 Ω/sq, 340 S/cm) of the GO film's sheet resistance (21.9 Ω/sq, 120 S/cm), which is a 2.83× higher electrical conductivity. These large differences in both mechanical and electrical properties between GO and L-GO films were mainly due to differences in the size of the flakes that composed each film: the average flake size in the L-GO film was about four times larger than that of GO films. In general, because graphenes are mechanically stiff flakes having high electrical conductivity, loads or electrons can be efficiently transferred within the graphene flakes. In contrast, at the grain boundaries, which are interconnecting regions between different graphene flakes, scattering of loads or electrons can occur due to the presence of structural discontinuity at these regions [30,31]. As the flake size decreases, the fraction of grain boundaries in the film increases, which can act as scattering centers of loads and electrons, degrading both mechanical and electrical properties. Therefore, the L-GO film showed better mechanical properties than the GO film due to less scattering of loads and electrons at grain boundaries. Compared to our reduced L-GO film, the state-of-the-art graphene film in which the GO flakes with a lateral size of 5 μm are strongly interconnected via physical crosslinking by evaporation of GO hydrogels showed better mechanical and electrical properties (ultimate tensile strength = 614 MPa and electrical conductivity = 802 S/cm) [32]. However, we expect our larger GO flakes will result in GO film with better mechanical and electrical properties if the interface between graphene flakes improves.

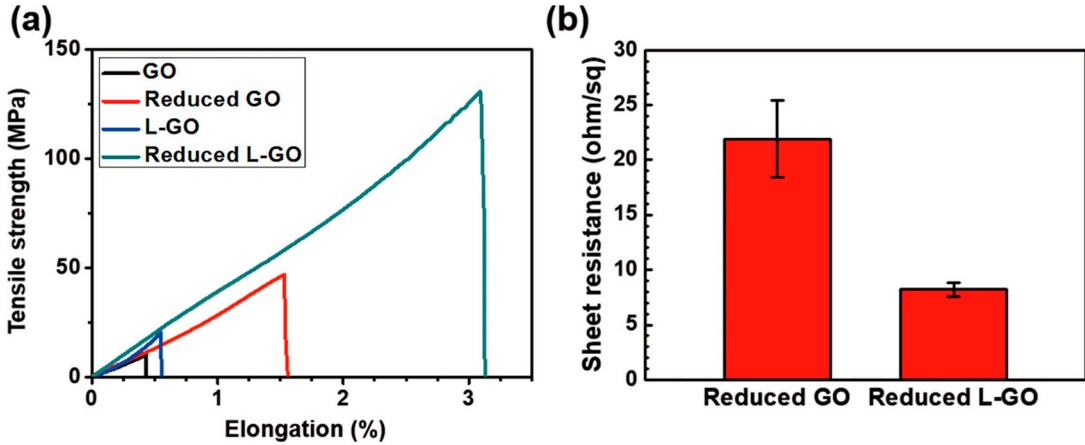

**Figure 5.** Mechanical strength and electronical conductivity of GO and L-GO: (**a**) strength–strain curve and (**b**) sheet resistance.

## 4. Conclusions

In summary, the size-dependent liquid crystal behavior of graphene oxides was studied. The GO suspension with larger-sized GO flakes (150 ± 29 μm) showed four times lower critical concentration for lamellar-like liquid crystal transition than the GO suspension with smaller-sized flakes (28 ± 24 μm); these results are in good agreement with the size-dependent liquid crystal behavior predicted by Onsager's theory. Graphene-based films made from the suspension with larger sized GO flakes showed much better mechanical and electrical properties due to the lower number of defects in the film compared to that of the film with smaller-sized GO flakes. We think that this study may provide a fundamental understanding of the liquid crystal behavior of GOs and will contribute to achieving graphene assembly with better properties at bulk scale.

**Author Contributions:** Conceptualization, S.R.; methodology, B.C. and S.J.; software, B.C.; validation, B.C. and S.J.; formal analysis, B.C., S.J., J.J. and G.J.; investigation, B.C., S.J., J.J. and G.J.; resources, S.R.; data curation, B.C. and S.J.; writing—original draft preparation, B.C. and S.J.; writing—review and editing, S.R.; visualization, B.J. and S.J.; supervision, S.R.; project administration, S.R.; funding acquisition, S.R. All authors have read and agreed to the published version of the manuscript.

**Funding:** This paper was supported by the research grant of the University of Suwon in 2018.

**Conflicts of Interest:** The authors declare no conflict of interest.

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
