# Peer review of "Size-Dependent Liquid Crystal Behavior of Graphene Oxides for Preparation of Highly Ordered Graphene-Based Films"

_applsci, doi:10.3390/app10165570_

Round 1

Reviewer 1 Report

This research reported the preparation of highly oriented graphene film from liquid crystal dispersion of GOs. The larger-sized GO flakes showed a more ordered state than the smaller sized GO flakes at the same concentrations. SEM, TEM, XPS, Raman, XRD, etc. were used as characterization methods. 

  1. The author called the 20mg/ml graphene gels - can the author provide their rheology performance? Their flow behavior using doctor blade may influence their orientations.
  2. The rL-GO showed much higher mechanical properties which may relate to the intrinsic properties of graphene or the interlayer interactions and defect distributions - how does the author explain the structure-property relationship?

Author Response

â—Ž Responses to Review 1

Comment 1: The author called the 20mg/ml graphene gels - can the author provide their rheology performance? Their flow behavior using doctor blade may influence their orientation.

As reviewer recommended, we added the rheology data (figure 2c) of both GO and L-GOs suspensions and the related discussion to the revised manuscript. The L-GO suspension exhibited shear-thinning behavior at lower concentration than the GO suspension. This shear thinning behavior suggests the transition of a nematic liquid crystal phase. We strongly agree that the orientation of GO flakes is highly affected by the fabrication condition of doctor blade such as gap size, speed, shear rate and the rheological behavior of GO suspensions. However, it is out of focus in the current manuscript, and we will discover these issues in the future research.

Our modification:

Page 4: The text “To further investigate the liquid crystal behavior of GO and L-GO suspension, we examined the shear rate dependencies of the viscosity for these suspensions at different concentration. The GO suspensions showed almost constant viscosity over a wide range of shear rates when their concentrations are 4 mg/ml and 12 mg/ml. However, when the concentration increased to 20 mg/ml, it showed a decrease in viscosity as the shear rate increased. Such viscosity change is known as shear-thinning which has been generally observed in nematic liquid crystal suspension.[26] For the L-GO suspension, clear shear-thinning behavior was observed even at the lowest concentration of 4 mg/ml, suggesting that L-GO suspensions can form the nematic liquid crystal phase at much lower concentration than GO suspension.”

Figure 2. Polarized optical microscopy (POM) images of (a) GO and (b) L-GO suspensions according to their concentrations. Insets show schematics of GO and L-GO flakes in suspensions for each POM image. Scale bars are 200 μm. (c) Plots of viscosity versus shear rate according to concentrations of GO and L-GO suspension.

Comment 2: The rL-GO showed much higher mechanical properties which may relate to the intrinsic properties of graphene or the interlayer interactions and defect distributions - how does the author explain the structure-property relationship?

In general, the mechanical properties of graphitic materials are highy affected by their crystallite domain size. When a load is applied to a graphtic material, the load is transferred within the crystallite domain, while it can be scattered at the boudary between domains. As the crystallite domaine size increases, the load trasfer within graphtic materials is improved, resulting in better mechanical properties. In the similar manner, the L-GO film showed better mecahnaical properties because it consisted of 4 times larger flakes than the flakes in GO films.

Pleae see our modication below:

Page 7: The text “These large differences in both mechanical and electrical properties between GO and L-GO films are mainly due to difference in the size of the flakes that make up each film: the average flake size in the L-GO film is about 4 times larger than that of GO films. In general, because graphenes are mechanically stiff flakes having high electrical conductivity, loads or electrons can be efficiently transferred within the graphene flakes. In contrast, at the grain boundaries, which are interconnecting regions between different graphene flakes, scattering of loads or electrons can occur due to the presence of structural discontinuity at these regions [31, 32]. As the flake size decreases, the fraction of grain boundaries in the film increases which can act as scattering centers of loads and electrons, degrading both mechanical and electrical properties. Therefore, the L-GO film showed better mechanical properties than the GO film due to less scattering of loads and electrons at grain boundaries. Compared to our reduced L-GO film, the state-of-the-art graphene film in which the GO flakes with a lateral size of 5 µm are strongly interconnected via physical crosslinking by evaporation of GO hydrogels showed better mechanical and electrical properties (ultimate tensile strength: 614 MPa, and electrical conductivity of 802 S/cm) [33]. However, we expect our larger GO flakes will result in GO film with better mechanical and electrical properties, if the interface between graphene flakes improves.”

Reviewer 2 Report

In the present manuscript, the authors reported the preparation and the characterization of oriented graphene films with a peculiar thickness taking advantages on the formation of a liquid crystalline phase in water dispersion. Chemical reduction by means of HI was eventually reported as well as final characterisations in terms of mechanical test and electrical conductivities. Notwithstanding the overall interest on the selected topic, the manuscript contains several shortcomings and issues to be addressed:

1) the introduction is too shorh, it's non enough to introduce the topic, the issues behind the formation of graphene oriented layers and the state of the art;

2) the experimental part is too poor as well: no information about the realisation of the mechanical experiments and the determination of the electrical conductivities is still missing as well. The concentration of HI is not reported for the chemical reduction of GO;

3) SEM are not enough. In order to determine the layer thickness, TEM should be reported;

4) the discussion of the XRD experiments and results are quite weird. One is expecting a short to low theta angles as the reduction of the layer thickness proceeds (Bragg's low). The authors reported an opposite behaviour.

5) the determination of the liquid-crystalline phase by only means POM is not enough. Please provide stronger evidences.

6) no discussion of the mechanical data and electrical conductivities were reported by authors in terms of the recent literature. The reader is not able to determine the importance of the data reported  

Author Response

â—Ž Responses to Review 2

Comment 1: the introduction is too short, it's not enough to introduce the topic, the issues behind the formation of graphene-oriented layers and the state of the art.

            We have modified introduction to explain the critical issue we dealt in our work. Please see our modification below.

            Our modification:

Page 1: The texts “To realize a bulk graphene assembly, graphene oxides (GOs) have been widely used as starting materials. Because GOs have many functional groups and can be dispersed in various solvents[5, 6], a bulk graphene assembly can be produced by various solution processes such as vacuum filtration[7], wet spinning[8], casting[9], and the electro-spray roll to roll process[10].

Currently, most graphene assemblies exhibit significantly lower properties than the monolayer graphene due to the presence of defects, which are mainly located in the interconnection regions between each graphenes. In order to obtain a graphene assembly with better properties, defects within assemblies should be reduced by realizing more ordered graphenes. Recently, a nematic liquid crystal behavior of GOs in which GO flakes are aligned in a colloidal dispersion has been reported[11, 12, 13]. By subsequent solution processing of such dispersion, a highly-ordered graphene assembly can be obtained[14, 15, 16].

It is known that liquid crystal behavior of GOs is highly affected by various factors such as concentration, lateral size and functional group of GO flakes[12, 13], but more systematic study is required to better understand of liquid crystal behavior of GOs and obtain a more ordered GO assembly.”

Comment 2: the experimental part is too poor as well: no information about the realization of the mechanical experiments and the determination of the electrical conductivities is still missing as well. The concentration of HI is not reported for the chemical reduction of GO.

We have added how we characterize mechanical and electrical properties of graphene-based films, and the concentration of HI with the references in the experimental section.

Page 3: The texts “The dried GO and L-GO films were reduced by HI [17, 18, 19]. GO or L-GO films were dipped in 10 ml of HI (55% in water) for 15 min”

and

The texts “Tensile strength was investigated by using a microforce testing machine (MicroTester 8848; Instron, Korea), and tensile tests were on rectangular samples with a 16 mm gauge length at a rate of 2.5 mm/min. The sheet resistance of films was measured with a 4-point probe system (CMT-SR1000N, Advanced instrument technology, Rep. of Korea. The cross-sectional area of the film was precisely evaluated from the cross-sectional SEM image using an image analysis program (Image J, NIH, USA). The electrical conductivity of the films was obtained by dividing the sheet resistance into the cross-sectional area.”

Comment 3: SEM are not enough. In order to determine the layer thickness, TEM should be reported.

We have added high resolution SEM images, which clearly showing the thickness of graphene-based films in the figure 3c.

Figure 3. (a) Schematic of preparation of graphene-based films from liquid crystal GO suspensions. Gray and red circles indicate carbon atoms and functional groups, respectively. (b) Photograph of GO and L-GO films before and after reduction. (c) Structural change in graphene-based films by HI reduction: schematic illustration (top), and cross-sectional SEM images (bottom). Scale bars are 100 μm, and insets are high magnification images of the regions marked with white rectangles. (d) XRD spectra of GO and L-GO films before and after HI reduction.

Comment 4: the discussion of the XRD experiments and results are quite weird. One is expecting a short to low theta angles as the reduction of the layer thickness proceeds (Bragg's low). The authors reported an opposite behavior.

“GO (and L-GO)” and “Reduced GO (and Reduced L-GO)” were marked in reverse. We apologize for this oversight and have corrected this in the revised manuscript. Please see the modified figure 3d.

Our modification:

Figure 3. (a) Schematic of preparation of graphene-based films from liquid crystal GO suspensions. Gray and red circles indicate carbon atoms and functional groups, respectively. (b) Photograph of GO and L-GO films before and after reduction. (c) Structural change in graphene-based films by HI reduction: schematic illustration (top), and cross-sectional SEM images (bottom). Scale bars are 100 μm, and insets are high magnification images of the regions marked with white rectangles. (d) XRD spectra of GO and L-GO films before and after HI reduction.

Comment 5: the determination of the liquid-crystalline phase by only means POM is not enough. Please provide stronger evidences.

As reviewer recommended, we added the rheology data (figure 2c) of both GO and L-GOs suspensions and the related discussion to the revised manuscript. The L-GO suspension exhibited shear-thinning behavior at lower concentration than the GO suspension. This shear thinning behavior suggests the transition of a nematic liquid crystal phase.

Our modification:

Page 4: The text “To further investigate the liquid crystal behavior of GO and L-GO suspension, we examined the shear rate dependencies of the viscosity for these suspensions at different concentration. The GO suspensions showed almost constant viscosity over a wide range of shear rates when their concentrations are 4 mg/ml and 12 mg/ml. However, when the concentration increased to 20 mg/ml, it showed a decrease in viscosity as the shear rate increased. Such viscosity change is known as shear-thinning which has been generally observed in nematic liquid crystal suspension.[26] For the L-GO suspension, clear shear-thinning behavior was observed even at the lowest concentration of 4 mg/ml, suggesting that L-GO suspensions can form the nematic liquid crystal phase at much lower concentration than GO suspension.”

Figure 2. Polarized optical microscopy (POM) images of (a) GO and (b) L-GO suspensions according to their concentrations. Insets show schematics of GO and L-GO flakes in suspensions for each POM image. Scale bars are 200 μm. (c) Plots of viscosity versus shear rate according to concentrations of GO and L-GO suspension.

Comment 6: no discussion of the mechanical data and electrical conductivities were reported by authors in terms of the recent literature. The reader is not able to determine the importance of the data reported.

            We compared the properties of state-of-the-art graphene films to our films. Please see below:

Page 7: Compared to our reduced L-GO film, the state-of-the-art graphene film in which the GO flakes with a lateral size of 5 µm are strongly interconnected via physical crosslinking by evaporation of GO hydrogels showed better mechanical and electrical properties (ultimate tensile strength: 614 MPa, and electrical conductivity of 802 S/cm) [33]. However, we expect our larger GO flakes will result in GO film with better mechanical and electrical properties, if the interface between graphene flakes improves.

Round 2

Reviewer 2 Report

The authors have substantially improved their manuscript, I suggest only to describe the viscosity experiments in the experimental part (instrument, method, etc...)

Author Response

We appreciate the comment by reviewer. Responses to all of the comments from the reviewer are below. In addition, our major changes were marked with red texts in the revised manuscript.

â—Ž Responses to Review

Comment 1: The authors have substantially improved their manuscript, I suggest only to describe the viscosity experiments in the experimental part (instrument, method, etc...)

            As reviewer suggested, we added instruments and method we used for viscosity experiments at the ‘Materials and Methods’ section. Please refer our modification below.

            Our modification:

Page 3: The texts “Polarizing microscopy (ECLIPSE LV100ND POL, Nikon, Japan) was used to observe the liquid crystal behaviors of GO and L-GO and their shear viscosity was measured by rheometer (MCR 300, Paar Physica, Austria). Before the measurement, pre-shear of 500/s was applied for 60 seconds to ensure homogeneous dispersion of suspension. Between shear rates ranging from 20 to 760/s, viscosity of each GO suspension was measured in a stepwise manner for 40 minutes.”